# Barriers and Facilitators to Smart Healthcare Adoption Among Chinese Patients with Cardiovascular Disease and Their Caregivers: A Qualitative Study

**DOI:** 10.3390/healthcare13222881

**Published:** 2025-11-12

**Authors:** Zhaoying Zhu, Siying Ji, Xinyue Shi, Shan Li, Ruonan Yang, Menghan Zhu, Yunying Hou

**Affiliations:** 1The First Affiliated Hospital of Soochow University, Suzhou 215006, China; 20235231006@stu.suda.edu.cn (Z.Z.); 20234231011@stu.suda.edu.cn (S.J.); 2School of Nursing, Suzhou Medical College of Soochow University, Suzhou 215006, China

**Keywords:** cardiovascular disease, smart healthcare, self-care ability, caregiving ability, qualitative research

## Abstract

**Objective:** This study aimed to explore the barriers to and factors influencing the adoption of smart healthcare among Chinese patients with cardiovascular disease (CVD) and their caregivers with medium or low levels of self-care or caregiving ability. **Methods:** Semi-structured interviews were conducted with fourteen patients with CVD and nine caregivers to determine their needs and suggestions regarding the adoption of smart healthcare in Suzhou, Jiangsu Province, China. **Results:** The interview data were analyzed using Colaizzi’s seven steps. Two major themes and eleven subthemes were extracted, including facilitating factors (motivation to use, learning and interactive engagement, knowledge translation, and psychological identity) and obstacles (insufficient ease of operation, risk to personal privacy, ambivalence about paying for knowledge, fear of physical and mental injury, mistrust of implementation personnel, and technical or functional limitations of equipment) to the adoption of smart healthcare. **Conclusions:** Our findings suggest that before implementing smart healthcare interventions for patients with CVD and their caregivers, it is necessary to fully assess their willingness; push precise content based on their learning, interaction, and psychological needs; and address the technical barriers and privacy protection to enhance their willingness to use the system.

## 1. Introduction

Cardiovascular disease (CVD) is the leading cause of death and a significant contributor to disability worldwide [1]. In China, approximately 330 million people suffer from CVD, which is the leading cause of death and a great disease burden among residents [2]. Studies have shown that patients with CVD generally experience a reduced quality of life [3,4] and increased levels of anxiety and depression [5,6]. Their caregivers often experience a reduced quality of life [7], increased financial and caregiving-related burdens [8], and negative emotions [9]. Based on the dyadic illness management theory, illness appraisal by the patient–caregiver dyad influences both parties’ engagement in management behaviors, which in turn may impact the dyad’s physical and psychological well-being [10]. Within the dyad of patients with CVD and their caregivers, the patient’s self-care and caregiver’s caregiving abilities are crucial for improving the health outcomes of both individuals [11,12,13]. However, this dyad continues to face prevalent self-care and caregiving inadequacies [7,14], highlighting the urgent need for effective support strategies.

Smart healthcare, an emerging approach used to address the challenges of chronic disease management, leverages 5G, the Internet of Things, and artificial intelligence (AI) technologies and can improve disease management efficiency [15]. These technologies can be utilized by healthcare providers (e.g., medical staff and institutions) to deliver more personalized and efficient diagnostic and treatment services [16] and by patients and their caregivers to enhance autonomous health management and access real-time support [15,17]. Furthermore, the successful application of remote monitoring and mobile health platforms in chronic disease contexts, such as diabetes and cancer, has strengthened patient self-management [18,19]. However, generalizing the application of smart healthcare to all chronic disease populations may result in overlooking the unique needs of patients with specific conditions. CVD is characterized by sudden onset, complex management, and a high demand for real-time monitoring and emergency responses [20,21]. This implies that the functional requirements of patients with CVD and their caregivers for smart health tools (e.g., data accuracy and emergency linkage) may differ significantly from those of other chronic disease patient groups.

Although previous studies have applied smart health interventions to CVD management and observed positive effects [22,23], the intervention impacts show considerable heterogeneity across different studies [23,24] and are often difficult to generalize due to limitations in study design, sample representativeness, etc. [24,25,26]. More importantly, most of the aforementioned studies have examined patients’ or caregivers’ perspectives in isolation, failing to consider the two as a dynamically interactive dyad. The absence of this dyadic perspective prevents a deep understanding of the similarities and differences in their experiences (e.g., patients may focus more on the health benefits, while caregivers may prioritize whether the tools alleviate their burden), consequently hindering the design of smart health solutions that truly and precisely meet the needs of these people.

Therefore, by addressing the identified research gaps, this study aims to explore the barriers to and facilitators of smart healthcare adoption among Chinese patients with CVD and their caregivers with moderate-to-low levels of self-care and caregiving abilities. Through semi-structured qualitative interviews, we sought to gain an in-depth understanding of the acceptance, experiences, needs, and expectations of smart healthcare within this specific population. These findings provide insights for smart health product designers, service providers, and healthcare policymakers, thereby promoting the innovation and development of smart healthcare service models. Ultimately, this study seeks to enhance the willingness of patients with CVD and their caregivers to use smart healthcare in disease management, with the goal of improving their disease management capabilities, and consequently the health outcomes of both these parties. Furthermore, this study provides evidence regarding the adaptability and feasibility of smart healthcare across diverse cultural, economic, and social contexts. This study is expected to offer insights into delivering more personalized and higher-quality medical services to patients with CVD and their caregivers, thereby providing a valuable reference for the global promotion and optimization of smart healthcare.

## 2. Materials and Methods

### 2.1. Design and Setting

This study employed a qualitative research design guided by a phenomenological methodology to explore the in-depth experiences of patients with CVD and their caregivers with smart healthcare. This study was conducted at the inpatient cardiology ward of The First Affiliated Hospital of Soochow University. A detailed Study Design Diagram is provided in Appendix A.

### 2.2. Sampling and Recruitment

This study used convenience sampling. As a non-probability sampling technique, convenience sampling involves selecting participants based on their accessibility and willingness to participate [27]. This approach was chosen for several reasons: (1) as a qualitative study, this research prioritized the richness of information over statistical representativeness; (2) this study was conducted at a single medical center, and the target population (hospitalized cardiology patients and their caregivers) presented specific challenges for recruitment, making convenience sampling a practical choice to efficiently access information-rich cases, while ensuring feasibility; and (3) this method is particularly suitable for exploratory and feasibility studies, aligning with the preliminary nature of this investigation. Compared to probability sampling methods, such as random sampling, convenience sampling offers advantages in terms of cost-effectiveness, time efficiency, and ease of implementation [28]. However, its limitation lies in the potential lack of generalizability to broader populations due to sampling bias [29]. Therefore, to maximize research quality, stringent quality control standards were implemented across all stages, including participant screening, recruitment, and interviewing, to ensure that all the participants could provide experiences highly relevant to the research questions. The overall participant recruitment process is illustrated in Figure 1, and the detailed inclusion and exclusion criteria are presented in Table 1.

Specifically, patients hospitalized in the cardiology department of the hospital between January and May 2025 and their primary family caregivers were selected as study participants. Recruitment was conducted by a researcher (the first author) who had undergone specialized training. The process began with the initial screening of potential participants through the electronic medical record system, followed by an adaptability assessment conducted by their attending physicians to evaluate their clinical stability and communication capacity. Eligible patients and their caregivers who passed the assessment were provided with detailed information regarding this study’s purpose, procedures, potential risks, and benefits. Subsequently, they were invited to sign written informed consent forms. During the study period, 30 CVD dyads were recruited. Among them, twenty-eight patients and twenty-nine caregivers completed Exercise of Self-Care Agency Scale (ESCA) and Caregiver Preparedness Scale (CPS) (two patients and one caregiver were excluded due to incomplete scale responses). Of the participants who completed the scales, 24 patients and 18 caregivers were identified as providing moderate-to-low levels of self-care or caregiving. Subsequently, in the interview phase, ten patients (eight refusals or dropouts, but two participated in pre-interviews) and nine caregivers (seven refusals or dropouts, but two participated in pre-interviews) were excluded. Consequently, a final sample of fourteen patients and nine corresponding caregivers completed the formal interviews.

Sample size determination adhered to the principle of data saturation in qualitative research, whereby recruitment ceased when no new themes or information emerged after the consecutive analysis of three interview transcripts [30]. Specifically, after the twenty-first interview (comprising data from thirteen patients and eight caregivers), no new categories emerged. To ensure the robustness of this conclusion, two additional interviews (with one patient and one caregiver) were conducted, confirming the recurrence of existing themes, without generating new insights.

### 2.3. Research Tools

#### 2.3.1. General Information Questionnaire

A questionnaire for the patients with CVD and their caregivers was developed based on a literature review and clinical practice experience. Each questionnaire consisted of two main sections. The first part was a general information form for the patient or caregiver. (1) The patient general information form included 10 items: gender, age, education level, financial pressure (self-rated on a three-level scale: “none”, “moderate”, or “severe”), place of residence, occupation, disease diagnosis, disease severity, course of the disease, and the Age-Adjusted Charlson Comorbidity Index (ACCI). Based on the ACCI score, the patients were categorized into four risk levels: low risk (0–2 points), moderate risk (3–5 points), high risk (6–8 points), and very high risk (≥9 points) [31]. (2) The caregiver general information form included 10 items: gender, age, education level, financial pressure, place of residence, occupation, the patient’s disease diagnosis, the patient’s disease severity, number of co-caregivers, and caregiving duration.

The second part consisted of scales measuring the self-care ability of patients and the caregiving ability of caregivers. (1) The Exercise of Self-Care Agency Scale (ESCA), designed and developed by the American scholars Kearney and Fleischer based on Orem’s self-care theory [32], was used to measure the patients’ self-care ability. The scale employs a Likert 5-point scoring method, with each item scored from 0 to 4 (“Very unlike me” = 0, “Somewhat unlike me” = 1, “Undecided” = 2, “Somewhat like me” = 3, and “Very like me” = 4). The scale comprises 43 items in total. Before calculation, the reverse-scored items needed to be converted from 0–4 to 4–0. The total ESCA score was then computed using the formula: $ESCA=\sum_{i=1}^{43}X_{i}$, where X_i_ represents the final score of the i-th item (i = 1, 2, 3, …, 43) and Σ denotes summation. The theoretical total score ranges from 0 to 172, with higher scores indicating stronger self-care ability. A score >66% (i.e., ≥114) indicates a high level, 33%–66% (i.e., 57–113) a medium level, and <33% (i.e., ≤56) a low level [32]. This study used the Chinese version of the ESCA [33], which reported a Cronbach’s α coefficient of 0.85–0.92, a test–retest reliability of 0.91, and a content validity of 1.0. (2) The Caregiver Preparedness Scale (CPS), developed by Archbold et al. [34], was used to measure caregivers’ ability. The scale also uses a Likert 5-point scoring method, with each item scored from 0 to 4 (“Strongly disagree” = 0, “Disagree” = 1, “Neutral” = 2, “Agree” = 3, and “Strongly agree” = 4). The total CPS score was then computed using the formula: $CPS=\sum_{j=1}^{8}Y_{j}$, where Y_j_ represents the score of the j-th item (j = 1, 2, 3, …, 8) and Σ denotes summation. The scale contains no reverse-scored items. The theoretical total score ranges from 0 to 32, with a higher score indicating better caregiving ability. A score >66% (i.e., ≥22) indicates a high level, 33%–66% (i.e., 11–21) a medium level, and <33% (i.e., ≤10) a low level. This study used the Chinese version of the CPS [35], which reported a Cronbach’s α coefficient of 0.925 and validity coefficients ranging from 0.706 to 0.839.

#### 2.3.2. Interview Outline

After reviewing the literature [36,37,38], engaging in discussions with the group members, and consulting with cardiology healthcare experts, an initial interview outline was developed. Subsequently, pre-interviews were conducted with two patients and two caregivers who met the scale score criteria. The guide was further refined based on the preliminary interview results to form the final version. Subsequently, one-on-one interviews were conducted with 22 patients and 16 caregivers. After excluding eight patients and seven caregivers who either declined to participate or dropped out midway, 23 formal interviews were completed. The specific content of the formal interview guide and the design intent behind each question are detailed in Appendix A.

### 2.4. Data Collection

Guided by the phenomenological methodology’s focus on in-depth lived experience, this study employed semi-structured interviews for data collection [39]. A semi-structured interview is an approach in which the researcher maintains a degree of control over the interview structure, while actively encouraging participants to engage and express themselves fully. Typically, the researcher prepares an interview guide in advance, which serves primarily as a prompt during the interview. The interviewer used the guide to pose questions but also encouraged the participants to raise their own issues and flexibly adjusted the procedure and content based on the flow of the conversation. This method combines the focus provided by a preset guide with the openness of flexible probing. This ensured comprehensive coverage of core topics (e.g., difficulties and facilitators in the usage experience), while allowing the participants the freedom to share their unique personal experiences and feelings, thereby generating rich and profound textual data. Compared to completely open, unstructured interviews or strictly closed, fully structured interviews, this approach offers a relatively consistent conversational framework with different participants without constraining their narratives within a rigid structure, which is conducive to subsequent cross-case comparisons and thematic analysis by the researcher.

The specific data collection procedure was as follows: First, prior to data collection, the researcher explained this study’s purpose, significance, procedures, and the participants’ rights in detail to each potential participant (CVD patient or their caregiver). Ample time was provided for questions to build a trusting relationship with the participants. After obtaining informed consent, the participants were assessed using either the ESCA or CPS. Subsequently, based on the assessment results, those identified as providing a moderate-to-low level of self-care or caregiving were invited to participate in the interview. The researcher scheduled a one-on-one, face-to-face, semi-structured interview in a quiet, undisturbed hospital conference room. During the interviews, the researcher paid close attention to nonverbal cues, such as facial expressions, body language, and tone of voice. This process was accompanied by continuous reflection, and all immediate impressions were recorded in memos. Each interview lasted approximately 20 to 40 min. All data collection was independently conducted by the primary researcher (a nursing science master’s student trained in qualitative research methods) between January and May 2025.

### 2.5. Data Analysis

To systematically implement the principles of phenomenological methodology and ensure the rigor of the research process, the interview data in this study were analyzed using the seven-step phenomenological analysis method proposed by Colaizzi [40]. This method aims to extract and understand the underlying structure of participants’ lived experiences from their specific descriptions of phenomena (i.e., the use of smart healthcare). The analysis in this study was reported with reference to the application example provided by Li et al. [41] and adhered to the Consolidated Criteria for Reporting Qualitative Studies (COREQ) [42] to ensure transparency.

The specific analytical steps were as follows: (1) repeatedly performing immersive reading of all interview transcripts to gain a holistic perception; (2) identifying and extracting significant statements relevant to the research phenomenon; (3) coding the statements and formulating meanings from them; (4) sorting and aggregating the formulated meanings into clusters of themes; (5) developing a detailed description of the theme clusters; (6) refining and integrating the themes to clarify their internal relationships, resulting in the final thematic framework (see Table 2); and (7) returning the analytical results to some of the respondents (seven patients and seven caregivers) for verification, enhancing the credibility of the findings through “member checking”. The entire analytical process was conducted independently by two researchers, with any discrepancies resolved through regular group discussions until a consensus was reached.

It is important to note that the subthemes under the theme “Barriers to adopting smart healthcare” were not further broken down into specific elements. This was primarily because during data analysis, the researchers found that the participants’ descriptions of barriers were often expressed as holistic perceptions or as core concerns. These subthemes were highly condensed and represented meaningful, stand-alone units of analysis that encapsulated the core experiences of the participants. Therefore, these subthemes were treated as the final “elements” in this study to more authentically reflect the overall narrative structure of the participants’ experiences regarding barriers.

### 2.6. Research Trustworthiness

To further ensure research quality, this study rigorously adhered to the criteria for establishing trustworthiness in qualitative research, as proposed by Lincoln and Guba [43]. Over the five-month study period, we collected and analyzed the scale data and in-depth interview materials from 23 participants, employing strategies of investigator and data triangulation to enhance credibility [44]. Specifically, two researchers independently coded and analyzed the data, and a consensus was reached through discussions with the research team. To address transferability, this study provided detailed descriptions of the research context and participant demographics, allowing readers to assess the findings’ contextual applicability. Furthermore, confirmability was ensured by maintaining a complete audit trail that included the raw data, analytical codes, and decision logs. The reliability of this study was supported by meticulous and standardized procedures for data collection and analysis.

## 3. Results

Fourteen patients with CVD and nine caregivers were interviewed. Of the fourteen patients with CVD, seven (50.0%) were male, and seven (50.0%) were female; their ages ranged from 49 to 83 (64.14 ± 11.2); eight (57.1%) had no economic stress, while six (42.9%) reported significant economic stress; their self-care ability score was 79.9 ± 18.4; the duration of disease was 1.9 ± 1.3 years; and the mean ACCI was 1.9 ± 0.7. Of the nine CVD caregivers, five (55.6%) were male, and four (44.4%) were female; their ages ranged from 21 to 68 (49.0 ± 18.9); five (55.6%) reported no economic stress, while four (44.4%) reported significant economic stress; their caregiving ability score was 16.1 ± 2.8; there were 1.1 ± 0.6 co-caregivers; and the duration of caregiving was 2.7 ± 2.0 years. The detailed information is presented in Table 3.

### 3.1. Thematic Framework of Smart Healthcare Use Experience

Through the analysis of the interview data, this study identified core themes reflecting the smart healthcare use experiences of the patients with CVD and their caregivers. As shown in Figure 2, experience is a dynamic outcome resulting from the interplay between facilitating factors and barriers. Facilitating factors enhance perceived value, self-efficacy, and continued use intention, whereas barriers may lead to use burnout, loss of trust, and discontinuation. The ultimate user experience and outcomes depend on the relative strengths of these two types of factors.

Furthermore, this study found that the patients and their caregivers, as a dyadic unit coping with the disease together, exhibited highly consistent views regarding their smart healthcare experience (highlighted in yellow in this figure). This reveals the common challenges and demands they face. Simultaneously, this figure clearly shows that the two groups have distinct priorities in their specific concerns, owing to their different roles. Specifically, the patients focus on acquiring disease-related knowledge, enhancing their self-identity, and ensuring their privacy and security. In contrast, the caregivers are more concerned with translating knowledge into concrete care actions to reduce their burdens.

### 3.2. Theme 1: Facilitators to Smart Healthcare Adoption

#### 3.2.1. Meeting Usage Motivation

The key to the widespread adoption of smart healthcare lies in its ability to meet users’ core needs. Patients with CVD and their caregivers face challenges such as disease management and access to medical resources. Therefore, there is a strong demand for efficient and convenient medical solutions.

##### Online Appointment Scheduling

Patients with CVD or caregivers can view doctors’ schedules via the app or the hospital’s website. This helps improve medical care efficiency, alleviate the pressure on hospitals, and reduce the risk of cross-infection, demonstrating the important role of smart healthcare in benefiting public health. A2: “I’ve had psoriasis for decades. Later, I consulted an online doctor about my condition. I think this platform provides good service in answering my questions.” B8: “Because my home is far from the hospital, I usually schedule appointments online and upload my reports for online consultations.”

##### Online Platform for Purchasing Medications

Users can purchase medicines via the online platform according to their individual needs and enjoy professional pharmacist services. This model effectively addresses the medication needs of patients, while also offering medication guidance and price transparency, significantly improving the accessibility of medicines. A10: “I sometimes watch health-related videos online or look up my symptoms to see what condition I might have. This helps me decide whether I can get medication from an online pharmacy.” B4: “He (the patient) recently caught a cold. I was afraid it might be the flu, so I checked the symptoms for a match and bought medicine online.”

##### Online Payment Platform

Online platforms provide patients with a convenient payment channel, allowing them to pay their expenses online. This service reduces the waiting times at service counters and is particularly suitable for returning patients and out-of-area medical treatment. It improves treatment efficiency and reduces the risk of large gatherings. A4: “I use a mobile phone to pay for various services at the hospital, such as ordering meals and paying additional fees.” B3: “I use a mobile phone to pay for my family’s medical insurance and hospital fees, which I find very convenient and time-saving.”

#### 3.2.2. Enabling Learning Participation

The ability to meet users’ learning and participation needs is a key factor in promoting smart healthcare. For patients with CVD and their caregivers, online guidance via smart devices enables them to master the use of remote monitoring equipment, learn health information, and obtain professional guidance.

##### Acquiring General Health Knowledge

Users can obtain knowledge about disease prevention and self-care through smart healthcare. These systems typically provide explanations, educational videos, and online question-and-answer services to help users master basic medical knowledge. A1: “I can find all the health-related information I need on my phone, such as methods for preventing seasonal diseases and tips for health preservation and wellness.” B3: “I usually use my phone to learn about health tips, such as which vegetables shouldn’t be refrigerated. It not only provides practical information but also reveals more unknown facts to me, like the high potassium content in bananas.”

##### Acquiring Knowledge of Disease

Users can obtain basic knowledge about the pathophysiological mechanisms of CVD through smart healthcare, such as the clinical manifestations and mechanisms of atherosclerosis formation. A3: “I think AI is effective. I describe my symptoms, and it provides a diagnosis and treatment plan in response. I believe it’s generally accurate and consistent with the diagnosis from a hospital doctor.” A11: “I usually use TikTok to learn about cardiovascular health, such as the mechanisms of disease and treatment methods, and I follow many experts in this field.”

##### Acquiring Guidance on Diet, Medication, and Exercise

Smart healthcare can generate dietary recommendations and medication reminders based on the individual needs of users. Users can utilize features, such as illustrated instructions and exercise monitoring, to assist with implementation. Specialist doctors will review the recommendations to help patients establish standardized self-care practices. A4: “It would be great if smart healthcare could provide me with targeted, long-term exercise guidance” A12: “If someone could remind me about things like diet and exercise regularly via video chat after I get home, I would be willing to participate in such an activity.”

##### Acquiring Guidance on Emergency Response and Chronic Disease Management

For example, emergency guidelines, such as Cardiopulmonary Resuscitation (CPR) and Automated External Defibrillator (AED), as well as smart early warning systems, can help patients and caregivers master key response skills and improve disease management capabilities. A13: “I once lost consciousness while working, and it still scares me. In a video I saw on my phone, the doctor recommends always carrying some emergency medicine and learning some first aid knowledge.” B1: “If possible, I would like to learn how to manage my mother’s condition using smart technology. My current knowledge is fragmented.”

#### 3.2.3. Enabling Interactive Participation

Smart healthcare can meet the need for user interaction and participation in the healthcare system. Patients with CVD and their caregivers can obtain personalized professional guidance through online channels, promptly resolve questions, significantly improve participation and compliance in health management, and achieve more effective disease control.

##### Implementing Dynamic Data Monitoring

Smart healthcare can track users’ dynamic data through wearable devices and remote monitoring systems, automatically identify abnormal fluctuations, and provide alerts to users. Meanwhile, the test data can be synchronized to the cloud for both doctors and patients to view the results. A1: “These devices allow me to check my blood pressure with a mobile phone, which is very convenient and quick.” B3: “My cholesterol is a bit high, so it would be great if the devices could monitor it in real time based on my needs. That way, I could adjust my lifestyle based on the data.”

##### Implementing Intelligent Health Reminders

Smart healthcare can leverage personalized data to deliver medication, follow-up appointment reminders, and alerts for abnormal indicators via app notifications, text messages, or smart devices. Additionally, the system integrates patient data to optimize reminder timing and content and supports synchronized notifications for caregivers to ensure critical health issues are not overlooked. A4: “I was hospitalized because of over-exercising. If my phone could alert me when I’ve reached my limit, I would know when to stop.” B7: “Sports watches can monitor your activity. For example, if I’ve been sitting for too long, my watch prompts me to get up and move, which I think is quite good.”

##### Implementing Diverse Health Management

Smart healthcare can provide users with comprehensive health management support. By identifying patients’ needs, corresponding services are pushed to achieve full-cycle health management, from disease treatment to humanistic care. A3: “When I’m unhappy, I talk to an AI because I think it can help me calm down.” B5: “If VR could be used in the treatment of CVD to display a 3D model of the heart, we could see more intuitively where the problem lies and understand our condition more clearly.”

##### Implementing Interactive Doctor–Patient Communication

Smart healthcare enables real-time online consultations. Doctors can access patients’ dynamic monitoring data and provide personalized recommendations with the help of AI-assisted diagnosis. Patients and caregivers can participate in communication simultaneously. This communication model can significantly improve the treatment efficiency. A8: “I hope that the data detected by the device can be shared directly with my doctor, enabling prompt follow-up.” B5: “It would be much more convenient than going to the hospital if doctors and nurses could use a WeChat group to communicate with and care for patients.”

#### 3.2.4. Promoting Knowledge Transformation

Smart healthcare can encourage users to turn what they learn into personal habits, thereby facilitating knowledge absorption. For patients with CVD and their caregivers, using smart healthcare for learning helps optimize disease management outcomes.

##### Implementing Knowledge Sharing

Smart healthcare can facilitate knowledge sharing. Patients and caregivers can share the latest clinical knowledge, promoting the development of knowledge among clinicians, researchers, and patients, and continuously improving the quality of medical care. A4: “My spouse and I often share health-related information so we can both learn new things.” B9: “I look up online what the surgery is actually like to help me make a decision.”

##### Implementing Knowledge Application

Smart healthcare can help users master knowledge systems and promote the conversion of knowledge into behavior. The system can generate easy-to-understand knowledge guidance based on the patient data. B2: “The knowledge I have gained from caring for my spouse can be put into practice. For example, I learned that it’s important to avoid eating too much salt.” B7: “Nowadays, AI is quite advanced. For example, if we provide it with a diet and an exercise schedule, it can generate a comprehensive guide. I usually follow these recommendations.”

##### Implementing Knowledge Storage

Smart healthcare can build a structured medical knowledge base to safely store and manage medical data and continuously collect the latest clinical guidelines, expert consensus, and typical case studies. A6: “I hope I can acquire health knowledge using a checklist or quiz format, so I could go back and review the questions I got wrong.” B4: “Yesterday, I learned doctors can easily access a patient’s medical information through the electronic medical records.”

##### Implementing Knowledge Feedback

Smart healthcare can use smart feedback to create a complete loop for learning health-related knowledge. The system can use big data to effectively improve users’ disease management capabilities. A10: “Our mobile phones are connected to big data nowadays. They recommend content we like, which keeps us engaged.” B7: “I think using these devices is very helpful for managing health knowledge and exercise routines.”

#### 3.2.5. Promoting Psychological Identification

Smart healthcare can satisfy users’ psychological needs. Through humanized interaction design, transparent data management, and reliable service commitments, the system can build users’ trust in technology.

##### Promoting Self-Identification

Smart healthcare enables users to visualize health improvement curves and reinforces their role as “health manager” through incentive mechanisms. This effectively enhances the patients’ sense of self-efficacy. A1: “I use my phone to check what foods I can eat, and I rarely get hospitalized for dietary restrictions. This has really boosted my confidence in managing my health.” A11: “I have taught my parents how to use mobile phones to look up questions, and they have gradually learned how to do it.”

##### Promoting Emotional Identification

Smart healthcare systems can recognize users’ emotions and provide emotional support and feedback. In addition, the system supports the establishment of emotional connections between different users, allowing users to feel understood and cared for. A5: “Since falling ill, I have been feeling somewhat uncomfortable and anxious. I wish it could talk to me and help me feel better.” B8: “My father (the patient) used to live alone. To check on him, I installed a camera. Now I can see him and talk to him whenever I want.”

##### Promoting Community Participation

Smart healthcare systems can promote experience sharing and emotional support among users by building social groups, forming social networks, and enhancing the continuity and positivity of disease management. A10: “I usually watch videos on apps and follow WeChat official accounts, which share useful health information.” B5: “I hope to create a support group for patients to enhance communication between patients and caregivers.”

### 3.3. Theme 2: Barriers to Smart Healthcare Adoption

#### 3.3.1. Insufficient Operational Convenience

During the use of smart healthcare, some patients and caregivers view the lack of ease of operation as an important obstacle. Complex menus, cumbersome registration processes, and a lack of age-friendly design reduce their willingness to use the product. A2: “I’m not very educated, so I find many complex functions difficult to operate. If they were simplified, they would be much easier.” B3: “We don’t have much experience with electronic devices. Therefore, some complex features are challenging for us to master.”

#### 3.3.2. Personal Privacy and Security Risks

Sensitive information may face many security risks during storage and transmission. This has led many patients to adopt a wait-and-see attitude toward smart healthcare. A5: “I think using my ID card is convenient, and I’m a bit worried that linking my personal information to my phone might cause a data leak.” A8: “I hope my privacy will be protected. Yesterday, I saw a report saying that AI could lead to privacy leaks and illegal use, which makes me very concerned.”

#### 3.3.3. Psychological Conflict over Knowledge Payment

Some smart healthcare is not free to use and requires users to pay for knowledge. However, the knowledge payment model often triggers conflicting emotions; they are eager to obtain professional health guidance but are reluctant to pay for it. A2: “I thought online consultations would be cheap, but they’re actually quite expensive. I also worry that it’s easy to be disappointed with the service.” B4: “Even though some health-related apps require payment, I found them not very useful, and therefore not worth the cost.”

#### 3.3.4. Concerns About Physical and Mental Safety

Patients with CVD and caregivers also lack trust in the reliability of diagnosis and treatment, worrying about the impact of radiation on health and the anxiety caused by continuous monitoring. A11: “WeChat has video calls now, which is great, but it also makes the elderly more vulnerable to scams. Some scammers are very convincing, so they need to be careful not to trust strangers.” B5: “I often wear a smartwatch, and I’ve seen online that using electronic devices for a long time might have health risks from radiation. I’m a bit concerned about this.”

#### 3.3.5. Distrust of Implementers

Users may have concerns about the quality of online medical consultations, for example, the effectiveness of online consultations, the authenticity of consultations with “well-known doctors”, and the quality of educational videos. A10: “I feel that some doctors offering online consultations might not be qualified. I think good doctors are usually very busy in the hospital and don’t have time for online consultations. So, I’m a little worried about how well they work in general.” B4: “I watched some videos where experts talked about how to treat diseases, but I found that some were misleading, others were unreliable, and some even tried to scam me into getting treatment I didn’t need.”

#### 3.3.6. Limitations of Device Technology or Functionality

The promotion of smart healthcare also faces practical obstacles in terms of equipment technology and functional limitations. Some technical shortcomings directly worsen users’ experiences and reduce trust. A1: “I think the data monitoring on these devices could provide more detail, for example by giving more accurate real-time feedback.” B8: “The system now combines online appointments with in-person visits, but schedules often get disrupted when doctors run behind. I hope technology can be developed to dynamically adjust appointment times and prevent these delays.”

## 4. Discussion

In this study, we conducted a comprehensive qualitative analysis to identify the enabling factors and barriers to smart healthcare. The results indicate that the enabling factors for smart healthcare primarily manifest in the following areas: the motivation to use, learning and interactive engagement, knowledge translation, and psychological identity. However, advances in smart healthcare are hampered by several obstacles, including insufficient ease of operation, risks to personal privacy, ambivalence about paying for knowledge, fear of physical and mental injury, mistrust of implementation personnel, and technical or functional limitations of equipment.

### 4.1. Adhering to the Parallel Principles of Precise Empowerment and Role-Specific Adaptation

This study demonstrates that patients and caregivers, as a dyadic unit collaboratively managing the disease, exhibit both a high degree of consensus and systematic differences rooted in their distinct roles in their experiences with smart healthcare. Their shared concerns regarding aspects such as device safety and operational convenience constitute universal requirements for the foundational design of smart healthcare. However, a deeper divergence lies in the fact that the patients’ experiences are centered on a sense of “control” over their disease, with their core demand being the enhancement of self-efficacy, whereas the caregivers’ experiences are more “task-oriented”, emphasizing the resultant reduction in their burden.

In light of this, when promoting smart healthcare for patients with CVD and caregivers, product designers must adhere to the principle of “Precise Empowerment and Role-Specific Adaptation”, moving beyond the traditional “single-user” model. (1) For the patient group, the interactive design of smart healthcare products should aim to strengthen their health autonomy. For instance, providing intuitive and trustworthy health data dashboards and personalized risk alerts enables them to clearly understand their own status, thereby boosting their confidence and ability to manage the disease. (2) For the caregiver group, the development of functional modules needs to focus on empowering their caregiving role. The emphasis should be on creating integrated tools for task management, automatic alerts for abnormal situations, and links to professional support resources. The primary goal is to effectively alleviate their physical and mental burden, rather than simply replacing their care work. (3) Furthermore, a collaborative mechanism between product designers and service providers should be established to ensure the “role customization” concept is embedded throughout the entire process. During the initial product design phase, methods such as user profiling and journey mapping should be used to accurately capture the differentiated workflows and pain points of patients and caregivers. In the service promotion stage, clear guidance for role switching or customized function packages should be provided to ensure both these parties can access what they need, achieving an experience optimization characterized by “harmony in diversity”.

### 4.2. Adherence to Technological Innovation and Traditional Safeguards in Parallel

Although the focal points of the patients and the caregivers differ, their demand for the integration of smart healthcare and traditional medical models is highly consistent. Based on the results of the interviews, some elderly participants and those with lower levels of education lacked interest in using such platforms, preferring the traditional healthcare models such as face-to-face consultations, or health education lectures for health management. This preference may be related to the lower level of understanding of technologies and lack of familiarity with operation. Although previous studies have confirmed the significant advantages of smart healthcare in terms of convenience, real-time access, and interactivity [42]. However, in the traditional medical model, patients prefer face-to-face consultations and communication with doctors to obtain more comprehensive and detailed explanations of their condition and psychological support, which helps improve treatment compliance and efficacy [45]. Additionally, doctors can gain a more comprehensive understanding of a patient’s condition through physical examinations [46], which can help address the shortcomings of smart healthcare. Previous studies have also shown that doctors and patients have recognition rates of 66.34% [46] and 68.1% [47], respectively, for the “online-offline integrated medical model”, suggesting that the two models should complement each other rather than replace each other.

Consequently, when promoting smart healthcare for patients with CVD and caregivers, significant emphasis should be placed on the connection and integration between smart and traditional healthcare. The results of this study indicate that establishing a hybrid service model that combines online and offline resources can better meet users’ needs for flexibility and convenience. Specifically, while preserving the advantages of traditional in-person diagnosis and treatment, smart healthcare service providers can further develop service systems that integrate functions, such as online follow-up consultations and remote monitoring. This provides patients with more diverse choices and enhances the overall service experience. Furthermore, collaboration between smart healthcare product designers and service providers is crucial. The design and validation of AI diagnostic tools should be fully integrated into actual clinical workflows. This ensures that while leveraging their efficient data processing capabilities, these tools also respect and support healthcare professionals’ clinical judgment, thereby enhancing the tools’ clinical applicability and acceptance. Secondly, regarding health information dissemination, service providers can establish a multi-channel communication system that combines online and offline methods. For example, alongside conducting online health education, activities such as community lectures and patient support groups should be organized. This approach helps broaden the reach of information, caters to the different information acquisition preferences of users, and thereby more effectively enhances the disease management capabilities of both patients and caregivers. Finally, at the level of health information management, service providers and product designers should, while promoting electronic health records, also retain and optimize the auxiliary management functions of paper-based medical records, creating a parallel and complementary information management model. This initiative not only helps ensure the accessibility and completeness of health information but also respects the usage habits of different user groups (especially digitally vulnerable groups), thereby strengthening their overall trust in the smart healthcare system.

### 4.3. Adherence to Service Optimization and Humanistic Care in Parallel

Similarly, optimizing the service experience is also key to meeting the common expectations of this dyadic user group. Some participants believed that using smart healthcare would present difficulties and challenges, such as incomprehensible professional terminology, dry and obscure educational content, complex interface design, and cumbersome functional modules. In addition, some participants expressed mixed feelings about using smart healthcare to improve emotional loneliness and anxiety. On the one hand, they believed AI communication can alleviate emotional burdens. On the other hand, they felt smart healthcare lacked warmth and human touch. Previous studies have shown that smart healthcare is not limited to interactions between users and service providers, but also encompass interactions between users and other tangible elements and intangible elements [48]. Based on this, interface designs that are friendly [49], effective interaction with family members and medical staff [50], and relief from emotional distress [51] are considered positive factors that promote the use of smart healthcare.

Therefore, when promoting smart healthcare for patients with CVD and caregivers, it is crucial to integrate technological application with humanistic care. At the technological design level, product developers should prioritize the needs of marginalized groups (e.g., the elderly) by optimizing interactive interfaces and enhancing features like voice recognition and multi-modal interaction, thereby substantially improving product accessibility and inclusivity. At the service delivery level, providers should leverage functions such as dynamic data monitoring, intelligent reminders, and remote management to achieve diversified and timely responses to users’ needs, ensuring service efficiency and accuracy. Regarding functional expansion, service providers and product designers should collaborate to incorporate modules for psychological counseling and personalized health advice into the system design, thereby enhancing the emotional support attributes of the service. At the clinical integration level, strengthened collaboration is also needed to combine data collected by smart devices with professional human judgment during diagnosis and treatment, ensuring that therapeutic plans are both scientifically precise and humane. Furthermore, product designers and community health workers could jointly explore using technology to strengthen the support network between patients and caregivers, for instance by integrating psychological support resources, shared responsibility mechanisms, and problem-solving tools to foster the healthy development of the dyadic relationship. Finally, service providers should enhance training for medical staff in communication skills and humanistic literacy, maintaining a balance between professional guidance and emotional support within technological applications, thus holistically increasing users’ trust and satisfaction with smart healthcare services.

### 4.4. Adherence to Equipment Safety and Multi-Party Collaboration in Parallel

Finally, privacy security and multi-stakeholder collaboration are crucial for building trust. Personal privacy protection [52], physical and mental well-being [53], and standardization of fees [54] are considered key factors promoting the adoption of smart healthcare among patients with CVD. In this study, many participants expressed concerns about potential personal information leaks when using smart healthcare. In addition, some participants expressed concerns about issues such as technical security deficiencies, insufficient professional competence of implementation personnel, and irregularities in fee-based services in smart healthcare, which often led them to worry about the effectiveness of safety protection during use.

However, the ability of patients with CVD and their caregivers to safely and effectively use smart healthcare services largely depends on synergistic cooperation among service providers, product designers, patients, caregivers, and other relevant stakeholders. Firstly, regarding data security and privacy protection, service providers should continuously strengthen technical safeguards and institutional norms. This includes (1) enhancing data encryption mechanisms, particularly improving device performance and data security in multi-party computing environments; (2) exploring advanced technologies like homomorphic encryption to bolster privacy protection within blockchain applications; and (3) conducting regular device performance evaluations, security protocol training, and establishing comprehensive quality supervision mechanisms and regulatory frameworks to systematically reduce security risks and strengthen user trust. Secondly, regarding enhancing the user’s experience, the establishment and optimization of multi-party collaboration mechanisms should be promoted. (1) Product designers should focus on device improvement and refinement, continuously breaking through technical barriers to achieve safe and efficient data monitoring and health promotion. (2) Service providers should improve hiring practices, strengthen qualification reviews, and enhance the skill levels and communication techniques of implementation personnel (e.g., platform customer service and medical staff). (3) Service providers and product designers should strengthen cooperation to optimize the user experience (for patients and caregivers), expand service channels, and conduct both online (e.g., sharing communities and discussion groups) and offline (e.g., public lectures and exchange workshops) communication activities to increase user engagement. (4) Healthcare policymakers should increase financial investment and policy support for product designers, encouraging relevant departments and medical institutions to conduct interventional research on telemedicine. (5) Furthermore, insurance companies should actively engage with service providers and product designers, fostering tripartite collaboration to continuously optimize the user claims process.

### 4.5. Limitations

This study qualitatively explored the smart healthcare experiences of Chinese patients with CVD and their caregivers providing low-to-moderate levels of self-care and caregiving. While the findings offer theoretical and practical insights, several limitations should be acknowledged. First, the sample was recruited from a single tertiary hospital in Suzhou, which may limit the generalizability of the results. Second, as a qualitative study, despite measures such as dual independent coding, team discussions, and participant feedback to enhance objectivity, data collection and interpretation may still be influenced by the researchers’ subjective perspectives. Third, reliance on one-time interviews prevented long-term tracking of participants’ attitudes and behavioral changes. Future research could adopt longitudinal designs or integrate ethnographic methods to better capture dynamic usage experiences. Finally, this study did not systematically track the patient–caregiver dyadic relationships during design, restricting the analysis of interactive experiences within pairs. Subsequent studies should emphasize dyad-specific tracking and employ corresponding analytical methods to more comprehensively reveal shared experiences and interaction mechanisms within care dyads.

## 5. Conclusions

Through in-depth analysis, this study has identified the core facilitating factors and barriers influencing the smart healthcare use experiences of Chinese patients with CVD and their caregivers. The findings indicate that users’ adoption intention fundamentally stems from whether the tools can meet their intrinsic needs in areas such as motivation for use, interactive participation, and psychological identification. Conversely, the primary obstacles are external factors, including technical ease of use, data security, and interpersonal trust. Therefore, the future development of smart healthcare must focus on resolving these key barriers while simultaneously strengthening its capacity to respond to users’ core needs. Achieving this “needs–barriers” balance is crucial for enhancing user trust, improving disease management capabilities, and ultimately promoting the sustainable development of smart healthcare.

## Figures and Tables

**Figure 1 healthcare-13-02881-f001:**
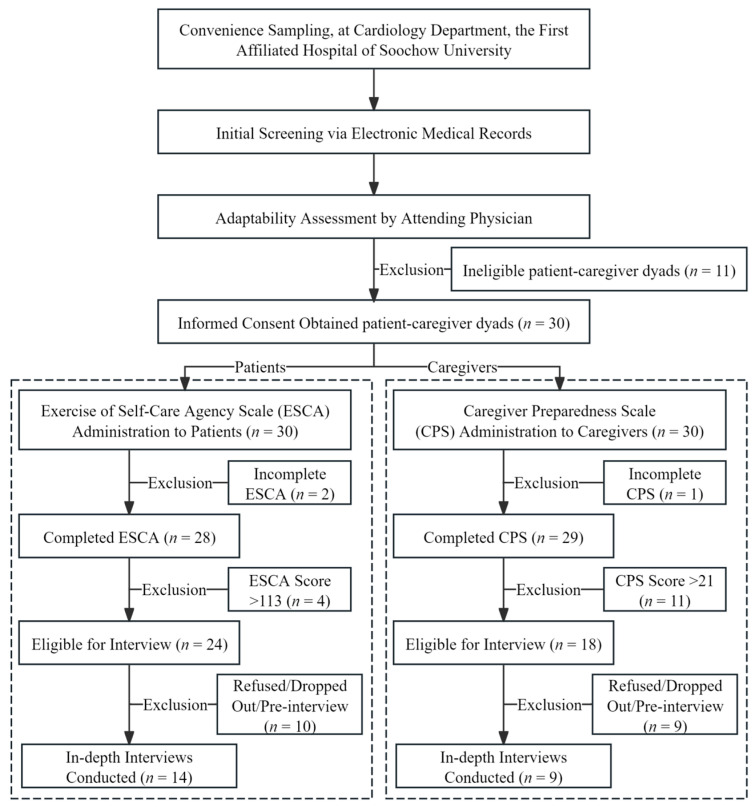
Flowchart of procedure.

**Figure 2 healthcare-13-02881-f002:**
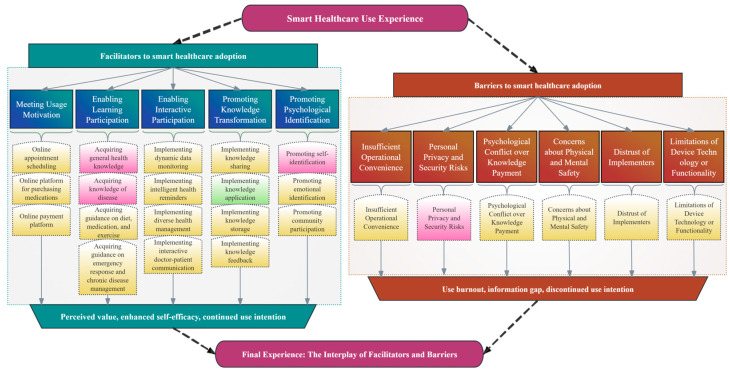
Thematic Framework of the Smart Healthcare Use Experience. The coding system for facilitating factors and barriers was derived deductively and inductively (Red: Patient perspectives; Green: Caregiver perspectives; Yellow: Shared perspectives).

**Table 1 healthcare-13-02881-t001:** Inclusion and Exclusion Criteria for Patients and Caregivers.

Criterion Type	Patients	Caregivers
Inclusion Criteria	1. Diagnosed with CVD by a cardiologist in a tertiary Grade A hospital; 2. Aged ≥ 18 years, conscious, with certain comprehension and verbal expression abilities; 3. Scored ≤ 113 points on the ESCA; 4. Voluntarily participated in this study and provided signed informed consent.	1. Primary family caregivers (including spouses, children, and relatives) providing the longest average daily care time (≥1 h/day); 2. Aged ≥ 18 years; 3. Scored ≤ 21 points on the CPS; 4. Voluntarily participated in the study.
Exclusion Criteria	1. Presence of cognitive, linguistic, auditory, or visual impairments that prevent effective communication; 2. Presence of severe comorbidities (e.g., severe heart, liver, or kidney dysfunction); 3. Experiencing severe psychiatric or physical symptoms during the interview period; 4. Inability to complete the interview for any reason.	1. Caregivers affected by severe personal illness or lack of self-care ability; 2. Presence of cognitive, linguistic, auditory, or visual impairments that prevent effective communication; 3. Hired, paid caregivers with a financial relationship; 4. Inability to complete the interview for any reason.

Note: CVD: cardiovascular disease; ESCA: Exercise of Self-Care Agency Scale (score range: 0–172; ≤113 indicates moderate-to-low self-care ability); CPS: Caregiver Preparedness Scale (score range: 0–32; ≤21 indicates moderate-to-low preparedness).

**Table 2 healthcare-13-02881-t002:** Table of Interview Themes.

Theme	Subthemes	Elements
Facilitators to smart healthcare adoption	Meeting Usage Motivation	Online appointment scheduling
		Online platform for purchasing medications
		Online payment platform
	Enabling Learning Participation	Acquiring general health knowledge
		Acquiring knowledge of cardiovascular pathophysiology and clinical basics
		Acquiring guidance on diet, medication, and exercise
		Acquiring guidance on emergency response and chronic disease management
	Enabling Interactive Participation	Implementing dynamic data monitoring
		Implementing intelligent health reminders
		Implementing diverse health management
		Implementing interactive doctor–patient communication
	Promoting Knowledge Transformation	Implementing knowledge sharing
		Implementing knowledge application
		Implementing knowledge storage
		Implementing knowledge feedback
	Promoting Psychological Identification	Promoting self-identification
		Promoting emotional identification
		Promoting community participation
Barriers to smart healthcare adoption	Insufficient Operational Convenience	——
	Personal Privacy and Security Risks	——
	Psychological Conflict over Knowledge Payment	——
	Concerns about Physical and Mental Safety	——
	Distrust of Implementers	——
	Limitations of Device Technology or Functionality	——

**Table 3 healthcare-13-02881-t003:** Participants’ characteristics.

**General information of patients with CVD (*n* = 14)**											
**No.**	**Gender**	**Age (years)**	**Education level**	**Economic stress**	**Place of residence**	**Occupation**	**Disease diagnosis**	**Severity of disease**	**ESCA**	**Duration of disease (years)**	**ACCI**
A1	Female	60	Junior high school	No	Urban	Retired	Coronary atherosclerotic heart disease	General	93	0.5	1
A2	Male	67	High school	No	Urban	Retired	Premature atrial contractions, premature ventricular contractions	General	102	1	2
A3	Male	71	Primary School	General	Rural	Farmer	Atrial fibrillation, atrial flutter	Serious	70	2	2
A4	Female	63	Secondary school	General	Urban	Cleaner	Coronary atherosclerotic heart disease	Serious	68	2	2
A5	Male	52	Primary School	No	Rural	Farmer	Hypertensive heart disease	General	86	1	2
A6	Male	52	High School	No	Urban	Driver	Exertional angina, ischemic heart disease	General	111	1	1
A7	Female	83	Primary School	General	Rural	Unemployed	Acute myocardial infarction	Serious	55	4	3
A8	Male	77	Primary School	No	Rural	Unemployed	Acute coronary syndrome, heart failure	Serious	56	3	2
A9	Female	67	Junior high school	General	Urban	Retired	Atrial fibrillation, coronary atherosclerotic heart disease	General	93	2	2
A10	Male	51	High School	No	Urban	Mechanical engineer	Dilated cardiomyopathy, heart failure	Serious	68	1	2
A11	Female	49	Junior high school	General	Urban	Unemployed person.	Coronary atherosclerotic heart disease	General	96	2	1
A12	Female	81	Elementary school	No	Rural	Retired	Third-degree atrioventricular block	Serious	54	5	3
A13	Female	58	Junior high school	No	Urban	Farmer	Paroxysmal atrial fibrillation, hypertrophic obstructive cardiomyopathy	General	87	1	1
A14	Male	67	Elementary School	General	Rural	Farmer	Atrial flutter, hypertension	General	80	1	2
**General information of caregivers (*n* = 9)**											
**No.**	**Gender**	**Age (years)**	**Education level**	**Economic stress**	**Place of residence**	**Occupation**	**Patient’s disease diagnosis**	**Patient’s severity of disease**	**CPS**	**Number of co-caregivers**	**Caregiving time (years)**
B1	Male	34	Bachelor’s degree	No	Urban	Self-employed	Heart Failure	Serious	19	0	2
B2	Female	68	Elementary School	General	Urban	Retired	Acute Coronary Syndrome	General	14	1	5
B3	Female	56	Junior high school	No	Rural	Farmer	Acute myocardial infarction	Serious	14	2	3
B4	Female	55	Primary School	General	Rural	Farmers	Atrial fibrillation	Serious	13	1	4
B5	Male	21	High School	No	Urban	Student	Coronary atherosclerotic heart disease	General	18	2	1
B6	Male	81	Junior high school	No	Rural	Retired	Coronary atherosclerotic heart disease	General	14	1	6
B7	Male	31	Secondary school	No	Urban	Factory worker	Heart failure	Serious	15	1	0.5
B8	Female	43	Bachelor’s Degree	No	Urban	Human resource management	Third-degree atrioventricular block	Serious	21	1	0.5
B9	Male	52	High School	General	Urban	Self-employed	Coronary atherosclerotic heart disease	General	17	1	2

Note: ESCA: Exercise of Self-Care Agency Scale (score range: 0–172; ≤113 indicates moderate-to-low self-care ability); ACCI: Age-Adjusted Charlson Comorbidity Index (score range: 0–37; a higher score indicates a greater comorbidity burden, categorized as low risk: 0–2, medium risk: 3–5, high risk: 6–8, very high risk: ≥9); CPS: Caregiver Preparedness Scale (score range: 0–32; ≤21 indicates moderate-to-low preparedness).

## Data Availability

The data presented in this study are available on request from the corresponding author due to privacy restrictions.

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
