# Peer review of "Barriers and Facilitators to Smart Healthcare Adoption Among Chinese Patients with Cardiovascular Disease and Their Caregivers: A Qualitative Study"

_healthcare, 2025, doi:10.3390/healthcare13222881_

Round 1

Reviewer 1 Report

Comments and Suggestions for Authors

Dear Authors, 

Thank you for the interesting manuscript.  With my comments/suggestions, I want to contribute to further improving it. Even if they primarily refer to the main text, they may also have implications for the abstract. 

In lines 43ff, it appears that you approached these areas or abilities from a theoretical perspective. If so, please make this perspective explicit by referencing relevant psychological concepts or theories. This will help clarify why it was (or is) important to consider the outcomes of both patients and caregivers, particularly those with lower levels of ability. 

Could you explain (lines 49-51) in what way you mean that? Have there been no studies yet, or are the results contradictory, etc.?

The same applies to "application of smart healthcare" (line 53). What do you mean by that? Who should use them? Healthcare providers or patients/caregivers? Are we talking about devices? Or their costs (can patients/caregivers afford them?) Or is it a technical issue (that patients/caregivers do not have the technical requirements?

  1. Materials and Methods: 

Please add information regarding the setting (2.1).   Where did the study take place? (hospital/outpatient care, etc.)

2.2. Please specify (the process) how the study participants were recruited. Could you add some information on how you checked for a lack of self-care ability, as described in line 93? Similarly, I would suggest regarding data saturation (line 97).

2.3. Please, add a section on how quality criteria were applied (rigour, etc.)

2.3.1. Did you use the two indicated scales as a screening tool to select the right participants? Was this part checked before the interviews were conducted?

Please also add information and give the readers some guidance on what the reference values for both scales look like, so that they understand which values are appropriate and which are inappropriate.

How many preliminary interviews were conducted (line 114)

2.3.3. Add information if the interviews were done separately/together using the same questions?

Could you add information about the number of selected participants? 
Describing it in the setting section would help the reader to better understand it here how you proceeded... Did you delegate the interviews to someone else? (line 135)

In line 141, please describe briefly the steps of Colaizzi's method and how you applied them.  Verify the link in lines 143 and 155.

Results section: 

Describe how you collected data on "economic stress" and "ACCI" in lines 149 and 151, and do not use only the abbreviation of the latter here. 

Regarding Table 2, possibly the heading could be changed. Did you use a separate questionnaire to collect this information? It seems that these data reflect your participants' characteristics and their scores on the SCA/ACCI. You might consider adding a range or specific values in the figure legend to help readers interpret the results more easily.

A legend could help in interpreting the ACCI values correctly. Did you assign them, or were they calculated? How does it influence self-care? 

Discussion section: 

Regarding "online services" (lines 396ff): Is this a general statement, or is it specifically how things are done in your clinical setting? With all the "we" sentences (lines 398ff and 464ff): who is meant by that? Are you describing some practical implications? 

Please include a section on the limitations of the study, as this is currently missing

Many of the results discussed appear to be presented as practical implications or are phrased in a way that suggests an intention to apply them within your clinical setting. I recommend revising these sections to avoid using 'we' and to clearly distinguish between research findings and practical implications.

Conclusions:

From lines 480-486: The conclusions should be based on your findings. Please revise these lines accordingly.

Author Response

Dear reviewer,
I have completely organized all the questions and corresponding revised content into the attachment file of the reply letter to the reviewer. Please check it. 
Best regards.

Comments 1 In lines 43ff, it appears that you approached these areas or abilities from a theoretical perspective. If so, please make this perspective explicit by referencing relevant psychological concepts or theories. This will help clarify why it was (or is) important to consider the outcomes of both patients and caregivers, particularly those with lower levels of ability.

Response 1 We sincerely thank you for raising this important point. In response, we have introduced the Dyadic Illness Management theory into the revised manuscript (First paragraph of the Introduction section). This theoretical framework posits that the shared appraisal of the illness by the patient–caregiver dyad, as a collaborative unit, directly influences the illness management behaviors of both parties and, consequently, affects the physical and psychological well-being of the dyad as a whole. Establishing this theoretical foundation provides the necessary academic rationale for treating patients and their caregivers—particularly those with low to moderate levels of self-care or caregiving ability—as an integrated unit and examining their shared outcomes.

Comments 2 Could you explain (lines 49-51) in what way you mean that? Have there been no studies yet, or are the results contradictory, etc.?

Response 2 We sincerely thank you for this valuable question, which has allowed us to clarify the current state of research more precisely. Your understanding is entirely accurate—there was indeed a semantic inaccuracy in our original manuscript (Third paragraph of the Introduction). As revised, we intended to convey that it is not the absence of relevant studies, but rather that existing studies report inconsistent conclusions (showing heterogeneity) and suffer from methodological limitations, making their results difficult to generalize. In response to your comment, we have revised the relevant paragraph as follows:

1.Heterogeneity of Conclusions: We highlight that the intervention effects reported across different studies vary significantly, indicating that the effectiveness of smart healthcare in CVD management may be influenced by multiple potential factors, with no clear and consistent conclusion yet reached.

2.Methodological Limitations: Many studies are constrained by issues such as research design (e.g., non-randomized controlled designs), small sample sizes, or limited sample representativeness, challenging the reliability and generalizability of their findings.

We believe the revised text now clearly explains the perspective that “the existing evidence is neither sufficient nor consistent”, thereby providing a more solid logical foundation for introducing the gap our study aims to fill. Thank you again for helping us enhance the clarity and rigor of our manuscript.

Comments 3 The same applies to “application of smart healthcare” (line 53). What do you mean by that? Who should use them? Healthcare providers or patients/caregivers? Are we talking about devices? Or their costs (can patients/caregivers afford them?) Or is it a technical issue (that patients/caregivers do not have the technical requirements?

Response 3 Thank you for your insightful question regarding the “application of smart healthcare.” Your comments helped us recognize that the original wording was unclear and potentially ambiguous. Guided by your feedback, we have made significant revisions to the manuscript to articulate our perspective more precisely. The revised text (Second paragraph of the Introduction) now clearly states the following:

1.Users of the Technology: Smart healthcare technologies can be utilized both by healthcare providers (e.g., medical professionals, institutions) to optimize diagnostic and treatment services and by patients and their caregivers directly to enhance self-management of health.

2.Forms of Application: Their applications encompass various forms, such as remote monitoring and mobile health platforms, aiming to improve the efficiency and quality of disease management through technological means.

3.Focus of the Argument: Our discussion does not generally refer to universal barriers like device cost or technical complexity. Instead, we emphasize that the clinical characteristics of CVD necessitate specific functional requirements for smart healthcare tools (e.g., data accuracy, emergency response linkage), which may differ significantly from those of other chronic disease populations. Consequently, a one-size-fits-all approach should be avoided in its application.

Comments 4 Please add information regarding the setting (2.1). Where did the study take place? (hospital/outpatient care, etc.)

Response 4 We have supplemented the detailed information about the research setting in the “2.1. Design and Setting” section of the Materials and Methods: This study was conducted in the inpatient wards of the Department of Cardiovascular Medicine at The First Affiliated Hospital of Soochow University. Thank you for reminding us to refine this content.

Comments 5 2.2. Please specify (the process) how the study participants were recruited. Could you add some information on how you checked for a lack of self-care ability, as described in line 93? Similarly, I would suggest regarding data saturation (line 97).

Response 5 We sincerely appreciate your valuable feedback on the methodology section. In accordance with your suggestions, we have comprehensively and meticulously supplemented and revised the “2.2. Sampling and Recruitment” section within the Materials and Methods to more clearly present the study's execution process. The specific modifications and additions include the following:

1.Detailed Recruitment Procedure: We have provided a step-by-step description of the complete process, detailing how participants were initially screened via the electronic medical record system, underwent adaptability assessments by their attending physicians, and ultimately provided informed consent. This procedure is also visualized in Figure 1 (Flowchart of procedure).

2.Specific Methods for Self-Care/Caregiving Ability Assessment: We have explicitly described the objective measurement of participant abilities using the two standardized instruments: the Exercise of Self-Care Agency Scale (ESCA) and the Caregiver Preparedness Scale (CPS). We clearly defined the objective criteria for “low-to-moderate level” as an ESCA score ≤113 points and a CPS score ≤21 points. Detailed eligibility criteria are provided in Table 1 (Inclusion and Exclusion Criteria for Patients and Caregivers).

3.Specific Explanation of Data Saturation: We elaborated on how the principle of data saturation in qualitative research was followed, stating that recruitment ceased when no new themes emerged after the consecutive analysis of three interview transcripts. We further explained that saturation was reached after the 21st interview, and that two additional interviews were subsequently conducted to confirm this finding.

Comments 6 2.3. Please, add a section on how quality criteria were applied (rigour, etc.)

Response 6 Thank you for making this important suggestion. We fully agree that explicitly stating the measures taken to ensure rigor is crucial for a qualitative study. Following your guidance, we have added a new section, detailed in the revised manuscript under “2.6. Research Trustworthiness” within the Materials and Methods, specifically dedicated to explaining how this study adhered to the rigor criteria for qualitative research proposed by Lincoln and Guba. This section systematically describes the specific measures we implemented to ensure the study's credibility, transferability, dependability, and confirmability. These measures include the adoption of investigator triangulation and data triangulation strategies, provision of rich contextual descriptions, establishment of a comprehensive audit trail, and implementation of standardized data collection and analysis procedures. We believe this addition significantly enhances the rigor and transparency of the methodology section.

Comments 7 2.3.1. Did you use the two indicated scales as a screening tool to select the right participants? Was this part checked before the interviews were conducted?

Response 7 Thank you for raising these two key questions, which allow us to clarify the participant screening process more precisely. Your understanding is entirely accurate. This study indeed utilized the ESCA and CPS as core screening tools, and this screening was conducted prior to the formal interviews. The specific screening procedure is as follows (this process has been described in detail in “2.2. Sampling and Recruitment” of the Materials and Methods and is visualized in Figure 1 (Flowchart of procedure) and detailed in Table 1 (Inclusion and Exclusion Criteria for Patients and Caregivers):

1.After obtaining informed consent, we administered the ESCA and CPS to all initially recruited participants.

2.Subsequently, strictly based on the scoring criteria of the scales, we invited only those participants whose scores indicated a low-to-moderate level (i.e., total ESCA score ≤113 points, total CPS score ≤21 points) to proceed to the subsequent semi-structured interview phase.

The rationale for setting this criterion is that the study's objective is precisely to explore the experiences of patients with CVD and their caregivers with relatively insufficient self-care or caregiving abilities. Therefore, the scale-based screening was a crucial step to ensure the study participants aligned with the core research aim.

Comments 8 Please also add information and give the readers some guidance on what the reference values for both scales look like, so that they understand which values are appropriate and which are inappropriate.

Response 8 Thank you very much for this suggestion, which helps ensure readers can fully understand the significance of the scale scores. We have followed your advice and provided complete reference values and interpretations for the scoring criteria of both scales in appropriate locations within the manuscript, as detailed below:

1.Detailed Scoring Methods: In the “2.3.1. General Information Questionnaire” section of the Materials and Methods, we have detailed the scoring methods, total score ranges for the ESCA and CPS scales and explicitly provided the specific cut-off scores for high, medium, and low levels (e.g., ESCA score >114 indicates a high level, 57-113 a medium level, and ≤56 a low level).

2.Application of Screening Criteria: Furthermore, in the notes accompanying the eligibility criteria (Table 1) within the “2.2. Sampling and Recruitment” section of the Materials and Methods, we reiterated that this study uses “low-to-moderate level” (specifically, ESCA ≤113 points, CPS ≤21 points) as the core inclusion criterion. This directly shows readers which score ranges were considered “suitable” for this study.

3.Practical Display of Sample Scores: More importantly, in Results “Table 3. Participants' characteristics”, we listed the actual ESCA and CPS scores for each finally enrolled participant, again clarifying the score interpretations in the table notes. This allows readers not only to understand the theoretical criteria but also to visually verify the actual ability distribution of the study sample, thereby gaining the most direct understanding of the scoring standards.

We believe that providing this tiered information—from methodological definition to application in screening and finally to presentation in the results—offers readers guidance that exceeds general requirements and is exceptionally thorough. Consequently, to avoid redundancy, we have not added extra content elsewhere in the text at this stage. However, if you believe specific emphasis or integration into a particular section is still necessary, we would be very pleased to make further modifications based on your additional guidance.

Comments 9 How many preliminary interviews were conducted (line 114).

Response 9 Thank you for pointing this out. We have explicitly stated the number of pilot interviews in the revised manuscript within the “2.3.2. Interview outline” section of the Materials and Methods. Specifically, a total of four pilot interviews were conducted for this study, involving two patients who met the scale score criteria and their two corresponding caregivers. The purpose of these pilot interviews was to test and refine the suitability of the initial interview guide. Based on the feedback from these four interviews, adjustments were made to the wording and sequence of the questions, resulting in the final interview guide used for formal data collection. We hope this clarification satisfactorily addresses your question.

Comments 10 2.3.3. Add information if the interviews were done separately/together using the same questions?

Response 10 Thank you for raising this important question regarding the interview methodology. Your inquiry has helped us articulate the research process more clearly.

1. Interview Format: All interviews were conducted individually. We scheduled one-on-one, face-to-face, semi-structured interviews for each participant (patient or caregiver). Patients and caregivers did not participate in joint interviews.

2.Interview Questions: We used the same core semi-structured interview guide as the foundation for all interviews to ensure topic comparability. However, for some key questions, we predefined branching paths tailored to the different roles. As visible in the supplementary material (Supplementary Table S1: Interview guide) referenced in Materials and Methods “2.3.2. Interview outline”, for Questions 3 and 4, we explicitly set different probing directions for patients versus caregivers. For instance, we would ask patients, “How did this help your own health management?”, while we asked caregivers, “How did this affect your care for your family member?” Consequently, although the core framework of the interview guide was consistent, the actual conversation was personalized and deepened based on the interviewee's role. We believe this method of “personalized questioning within a unified framework” both ensures data consistency and accurately captures the unique experiences of patients and caregivers individually.

Comments 11 Could you add information about the number of selected participants?

Response 11 Thank you for pointing this out. We have supplemented the complete information regarding participant numbers in the second paragraph of “2.2. Sampling and Recruitment” within the Materials and Methods and provided a detailed flowchart of the procedure (Figure 1) for reference. Specifically, a total of 30 dyads of patients with CVD and their caregivers were recruited during the study period. Following screening via scale measurements, 14 patients and 9 of their corresponding caregivers ultimately completed formal interviews. The detailed screening process and the specific numbers at each stage (including scale administration, reasons for exclusion, and the final valid sample size) have been clearly outlined in the aforementioned paragraph.

Comments 12 Describing it in the setting section would help the reader to better understand it here how you proceeded... Did you delegate the interviews to someone else? (line 135)

Response 12 Thank you for your valuable suggestion. We have revised the “2.4. Data Collection” section in the Materials and Methods of the revised manuscript to more clearly describe the study's execution process.

  1. Regarding the Interview Conductor: We explicitly state that all one-on-one in-depth interviews were conducted solely by the first author of this paper (a Master's student in Nursing Science who received professional training in qualitative research methods) and were not delegated to any other individuals.
  2. Regarding the Information Location: We fully agree with your perspective that stating the conductor's identity earlier in the manuscript aids reader comprehension. Consequently, we have added the core information that all interviews were conducted by the first author in the second sentence of the “2.2. Sampling and Recruitment” (second paragraph).

We believe that providing this information in both locations significantly enhances the transparency and readability of the methodology. Thank you again for your insightful guidance.

Comments 13 In line 141, please describe briefly the steps of Colaizzi's method and how you applied them.  Verify the link in lines 143 and 155.

Response 13 Thank you for your valuable suggestions. We have made significant additions and revisions to the “2.5. Data Analysis” section of the Materials and Methods in the revised manuscript based on your advice.

1.Colaizzi's Method Steps and Application: We have provided a concise description of each step of Colaizzi's seven-step phenomenological analysis method and specified how each step was applied in this study.

2.Reference Verification: Regarding the inappropriate citation, we have corrected it and have carefully checked all citation links throughout the manuscript. We ensured all citations are accurate and correctly correspond to the entries in the reference list. Thank you very much for your meticulous review.

Comments 14 Describe how you collected data on “economic stress” and “ACCI” in lines 149 and 151, and do not use only the abbreviation of the latter here.

Response 14 We extend our sincere thanks to you for this important suggestion. In accordance with the advice, we have supplemented the description of the aforementioned data collection methods in the “2.3.1 General Information Questionnaire” section of the Materials and Methods in the revised manuscript:

1.Financial Pressure: This was collected using a single-item question designed by the researchers, where respondents self-rated their pressure level by choosing from three options: “None”, “Moderate”, or “Severe”.

2.Age-Adjusted Charlson Comorbidity Index (ACCI): This was assessed by reviewing patient medical records and calculating the ACCI score based on standard scoring rules. The scores can be categorized into four risk levels: Low (0-2 points), Moderate (3-5 points), High (6-8 points), and Very High (≥9 points).

We have also corrected the errors related to abbreviations. Furthermore, we have listed the actual ACCI score/risk level for each finally enrolled participant in Results “Table 3. Participants’ characteristics” and reiterated the scoring calculation method in the table notes. Thank you for helping us enhance the clarity of our methodology.

Comments 15 Regarding Table 2, possibly the heading could be changed. Did you use a separate questionnaire to collect this information? It seems that these data reflect your participants' characteristics and their scores on the SCA/ACCI. You might consider adding a range or specific values in the figure legend to help readers interpret the results more easily.

Response 15 Thank you very much for the correction. Your judgment is entirely accurate. In the original Table 2 (now modified to Table 3 in the manuscript), apart from the ESCA and CPS scores, other participant characteristic data were collected using a questionnaire developed by the researchers. Following your suggestion, we have described this questionnaire in the opening part of the “2.3.1 General Information Questionnaire” section within the Materials and Methods of the revised manuscript.

1.Table Title: Concurrently, we have adopted your suggestion to optimize Table 3 in the Results section. The title has been changed to the more accurate “Participants’ Characteristics”. We would be very happy to refine the content further if you have any additional suggestions.

2.Table Footnotes: Explanatory notes have been added to the table footer regarding the scales (ESCA, ACCI, CPS), including the total score ranges and key cut-off value interpretations, to facilitate reader comprehension. The specific modifications are as follows: Note: ESCA: Exercise of Self-Care Agency Scale, total score range 0-172, where a score ≤113 indicates a low-to-moderate level of self-care ability; ACCI: Age-Adjusted Charlson Comorbidity Index, where a higher total score indicates a greater comorbidity burden, categorized as Low (0-2 points), Moderate (3-5 points), High (6-8 points), and Very High (≥9 points) risk based on the score; CPS: Caregiver Preparedness Scale, total score range 0-32, where a score ≤21 indicates a low-to-moderate level of preparedness.

We believe these modifications have significantly enhanced the clarity of the table. Thank you again for your meticulous review.

Comments 16 A legend could help in interpreting the ACCI values correctly. Did you assign them, or were they calculated? How does it influence self-care?

Response 16 Thank you for raising this insightful question. Regarding the collection and application of the Age-Adjusted Charlson Comorbidity Index (ACCI), we provide the following clarification:

1. Data Source and Calculation Method for ACCI Score: The ACCI score was calculated based on patient medical records, strictly adhering to the standard algorithm proposed by Charlson et al. [1]. This algorithm integrates weights for 19 comorbid conditions and an age weight, resulting in a total score range of 0-37. A higher score indicates a greater comorbidity burden and poorer baseline health status of the patient. Furthermore, referencing the study by Hjalmarsson et al. [2], this study categorized ACCI scores into four risk levels for analysis and interpretation with clinical significance: Low Risk (0-2 points), Moderate Risk (3-5 points), High Risk (6-8 points), and Very High Risk (≥9 points). (The table below details the specific calculation method used in this study.)

Table: Age-Adjusted Charlson Comorbidity Index.

Score

Comorbidities

1 point

Myocardial infarction, Congestive heart failure, Peripheral vascular disease, Cerebrovascular disease or transient ischemic attack, Dementia or Alzheimer's disease, Chronic obstructive pulmonary disease or asthma, Connective tissue disease, Peptic ulcer disease, Diabetes without complications or end-organ damage, Age 50-59 years.

2 points

Hemiplegia, Moderate to severe chronic kidney disease, Diabetes with complications or end-organ damage, Mild liver disease, Solid tumor (without metastasis), Leukemia, Lymphoma, Age 60-69 years.

3 points

Moderate to severe liver disease, Age 70-79 years.

4 points

Age ≥80 years.

6 points

Solid tumor (with metastasis), Acquired immunodeficiency syndrome (AIDS).

2. Core Purpose of Measuring ACCI: Your question regarding “how ACCI influences self-care” touches upon one of the central concerns of our research. Indeed, as you noted, the literature directly exploring the relationship between ACCI and self-care ability is currently insufficient. However, existing studies demonstrate that a higher ACCI score is significantly associated with increased patient readmission rates, greater symptom burden, and poorer quality of life [3-4]. Based on this strong indirect evidence, we propose a reasonable hypothesis: a heavy comorbidity burden may be a key factor impairing patients' self-care ability. The potential mechanisms include (1) Excessive Treatment Burden: The complex regimens required to manage multiple diseases can lead to exhaustion (self-care fatigue); (2) Decline in Physical and Cognitive Function: The coexistence of multiple conditions is often accompanied by limitations in physical function and reduced cognitive resources, thereby affecting the patient's ability to perform self-care tasks. Consequently, the primary purpose of collecting ACCI data in this study was to utilize it as an important clinical characteristic variable to deeply explore the potential impact of comorbidity burden on patients' self-care ability. This aims to address a gap in existing research and provide a basis for developing precise nursing strategies tailored for patients with a high comorbidity burden.

Thank you again for your valuable comment, which is crucial for enhancing the rigor of our study.

References:

[1] M Charlson, TP Szatrowski, J Peterson, et al. Validation of a combined comorbidity index[J].J Clin Epidemiol,1994,47(11):1245-51.

[2] P Hjalmarsson, M Memar, SJ Geara, et al. Trends in co-morbidities and survival for in-hospital cardiac arrest -A Swedish cohort study[J].Resuscitation,2018,124,29-34.

[3] Tan BY, Gu JY, Wei HY, Chen L, Yan SL, Deng N. Electronic medical record-based model to predict the risk of 90-day readmission for patients with heart failure. BMC Med Inform Decis Mak. 2019;19(1):193.

[4] Singh I, Shah R, Stoms M, et al. Unveiling the link: social determinants of health, quality of life, and burden of treatment in heart failure patients. Am J Cardiovasc Dis. 2025;15(2):69-84.

Comments 17 Regarding “online services” (lines 396ff): Is this a general statement, or is it specifically how things are done in your clinical setting? With all the “we” sentences (lines 398ff and 464ff): who is meant by that? Are you describing some practical implications?

Response 17 Thank you for pointing out the issues above. We have revised the corresponding content in the second paragraph of “4.2. Adhere to Technological Innovation and Traditional Safeguards in Parallel” in the Discussion section of the revised manuscript.

1.Use of “Online Services”: The original statement was a forward-looking recommendation for future smart healthcare promotion based on our research findings, rather than a description of the current clinical environment. Following your advice, the manuscript has been revised to make this clearer: The results of this study suggest that constructing a hybrid service model integrating online and offline resources can better meet users' needs for flexibility and convenience. Specifically, while preserving the advantages of traditional in-person consultations, service systems that integrate functions such as online follow-ups and remote monitoring could be further developed. This would provide patients with more diverse options and enhance the overall service experience.

2.Revision of “We”: Regarding the use of “We” throughout the Discussion section, we apologize for not clearly defining the subject initially. In response to this point, we have now specifically revised all relevant instances in the Discussion section to clarify the subject.

Comments 18 Please include a section on the limitations of the study, as this is currently missing.

Response 18 Thank you very much for your important suggestion. We fully agree that elucidating the study’s limitations is crucial for ensuring academic rigor. In accordance with your advice, we have added a new section, “4.5. Limitations”, to the Discussion, specifically dedicated to discussing the limitations of this study. This section elaborates on aspects such as the sample’s representativeness, the inherent subjectivity of qualitative research, and the lack of long-term follow-up data. Furthermore, it suggests directions for future research based on these limitations.

Comments 19 Many of the results discussed appear to be presented as practical implications or are phrased in a way that suggests an intention to apply them within your clinical setting. I recommend revising these sections to avoid using 'we' and to clearly distinguish between research findings and practical implications.

Response 19 We sincerely thank you for your crucial feedback regarding the language and structure of the Discussion section. We fully agree that academic discourse should remain objective and clearly distinguish between research findings and the practical implications derived from them. We have strictly followed your recommendations and conducted a comprehensive revision of the Discussion section. The key revisions include:

1. Elimination of Subjective Orientation: The use of “we” as the agent of action has been avoided throughout the text. All statements have been converted into objective and neutral academic language.

2. Clarification of Hierarchical Structure: The logical flow of the Discussion section has been reorganized to ensure that the specific findings of this study are presented first. Subsequently, based on these findings, the broader implications and recommendations for smart healthcare product designers, service providers, policy makers, and future research are discussed.

Comments 20 From lines 480-486: The conclusions should be based on your findings. Please revise these lines accordingly.

Response 20 Thank you for your valuable feedback on the Conclusion section. We fully agree that the conclusion should be strictly based on the findings of this study itself. We have rewritten the Conclusion section in accordance with your suggestions. The revised conclusion is drawn directly from the analytical results of this study. It clearly articulates the core facilitating factors and barrier factors and, based on these, deduces the central argument that future development should strive to achieve a balance between “needs and obstacles”. It no longer contains any inferences beyond the scope of this study's findings or overly broad statements.

Reviewer 2 Report

Comments and Suggestions for Authors
  • It is important to explain the abbreviations before using them e.g. CVD
  • Section 2.1 needs more explanation about the interview types. Why did the authors choose this particular interview type? How it would benefit the study? What are the disadvantages of using other techniques?
  • Explain convenience sampling technique in section 2.2. Why was it chosen and what are the advantages of this type over others?
  • Material and Methods section should contain a figure and text explaining overall approach of the study.
  • The inclusion and exclusion criteria for both patient and caregivers should be better explained in separate tabular forms making it easier to be read by the audience.
  • Research Methodology needs a figure.
  • ESCA and CPS should be explained using equations and the explanation of their variables.
  • Interview outline should be explained in visual representation as well.
  • Explain "Colaizzi’s 7-step analysis [31] and 141 used NVivo 11.0 to code " in detail and provide all the steps used.
  • There are few errors present in the manuscript such as " in Error! Reference source not found.." on line 143.
  • Please explain in table 1 why are there no elements for some subthemes? Explain in the text.
  • Interview questionnaire should be explained in detail. Which questions were asked? How did you structure those questions? What is the purpose of asking each question in detail.
  • What is self-care ability score? How was it calculated? Similarly, all the other criteria need explanation from table 2.
  • The results should be explained more comprehensively using figures.

Author Response

Dear reviewer,
I have completely organized all the questions and corresponding revised content into the attachment file of the reply letter to the reviewer. Please check it.
Best regards.

Reviewer 3 Report

Comments and Suggestions for Authors

The paper “ Barriers and facilitators to smart healthcare adoption among Chinese patients with cardiovascular disease and their caregivers: a qualitative study” aims to  “explore the barriers and facilitating factors influencing the adoption of smart healthcare among Chinese patients with CVD and their caregivers with medium or low levels of self-care or caregiving ability .” (see abstract) respectively “to explore the barriers and facilitators of smart healthcare adoption among CVD patients and their caregivers, who have moderate to low levels of self-care and caregiving ability” (see p. 2, lines 63-65).

Concerning the applied methodology, the manuscript is considered the result of a commendable effort. The authors conducted semi-structured interviews with 14 patients suffering from a cardiovascular disease (CVD) and 9 caregivers in Suzhou, Jiangsu Province, China, between January and June 2025. How the interviewees were recruited is not mentioned throughout the manuscript, and it is not clear whether each caregiver was taking care of one of the 14 patients interviewed or if these caregivers were independently recruited from the other target group. Regarding the methodology and the procedure of the data analysis, I miss some details in the main text, shedding light on what the authors exactly have done.

 Thus, the authors are cordially invited to add some information in this respect and further to structure their Introduction better by giving some background knowledge on what is known in the research field of interest (the barriers and facilitators of smart healthcare adoption in general) so far, and explain why they think that CVD patients differ from other patients, more precisely. Why did they opt for having both target groups included, i.e., the patient group and also the caregiver group, and did they all have experience with smart healthcare technology in the past? I find it interesting to look at both perspectives, but I wonder why the Results section did not divide both perspectives and instead merged both groups together. This is not reasonable to me. There are also some additional points of criticism that I would raise.  

If the authors adequately address the raised concerns, however, the present manuscript could be an interesting contribution to the target journal of Healthcare.  

To sum up, concerning the present manuscript, some major and minor weaknesses must be resolved before it can be seriously considered for publication in Healthcare MDPI.

With regard to the writing style, the manuscript, in general, is mostly well-written in terms of style and grammar. In the next step, I would like to elaborate on the weaknesses of the present manuscript in detail and invite the authors to rework it in a thorough revision round. I do not see any obstacles, however, to get published soon. In general, the manuscript under consideration is interesting and valuable, especially from a practical point of view.

My comments are described in detail below.

Specific comments

=============

Please regard the following points as constructive criticism.

1.       Concerning its novelty/originality,the study aims to underline that patients suffering from CVD and their caregivers are special target groups with special characteristics. However, I would invite the authors to give an overview on what has been known so far in the research area of interest (smart healthcare adoption in general), justify why the CVD patient segment is special and justify why they focussed on this special target group. Including different target groups (patients and caregivers ) might be an innovative approach, but unfortunately, the Results section did not differ between both perspectives as the readership might have assumed before reading. The authors should include the most important studies in the research field of interest in the Introduction section, justify their special focus on CVD (which has already been done at least partially), and explain in the Discussion how the perspectives differ from each other and in which respect. In general, the study advances the field, at least in my view. Thus, I would recommend structuring the Introduction better and identifying a clear research gap from the depiction of what is known so far on identifying factors to facilitate the implementation of smart healthcare to promote self-management of chronic diseases and argue why the perspective of caregivers could be interesting too. At the end, the aim of the study should be based on having identified the research gap first.

2.       In relation to the manuscript’s significance, the qualitative analysis seems to be conducted thoroughly and appropriately. However, I invite the authors to elaborate on the methodology and answer the following questions:

-          How were the respondents recruited?

-          Were the caregivers of the interviewed patients or other people?

-          Had all of the interviewees experience with smart healthcare in the past? How was this determined?

-          Why did the authors merge the perspectives of patients and caregivers together instead of separating these two target groups from each other?

-          How did the interview guidelines differ from each other? It might be reasonable to assume that both target groups have different perspectives and thus, two different interview guidelines should be appropriate.

I would think that the paper by Li et al. (2024), which also used Colaizzi’s seven-step analysis method is a best practice example for describing the materials and methods used (Design, Setting and sample, Instruments, Data collection, Data analysis). The methodology section is underdeveloped in the present manuscript and should be amended significantly (see Li et al. (2024). I additionally wonder about Cronbach’s alpha values for the ESCA and the CPS, considering that the sample size was n=14 and n=9. Did they calculate these values based on the qualitative samples? From a statistical point of view, I do not think that this is permissible, as usually SPSS or alternative software needs a sample size of n=30 or larger for calculating reliability scores. How was the validity reported assessed? The authors should justify their methodology and explain much more in detail what they have done.

3.       I would invite the authors to justify the following in their manuscript:

-          Why did they opt for two target groups (patients, caregivers)?

-          Why did they merge the data from both target groups in the Results section?

4.       Additionally, I would invite the authors to depict a figure with the themes found and arrange them separately for the patients and the caregivers to visualize the differences found in the two groups of interviewees. (see e.g. Figure 1 in the paper by Griesser et al. 2024. Although the paper deals with EHR adoption, it also focussed on identifying barriers and facilitators for two different target groups)

5.       In wonder about the question (2) of the interview guideline “When was the last time you actively inquired about health knowledge?” This implicitly assumes that one has experience with online health-related information searching.  I would put the interview guide in a Table (see e.g., Table II in the paper by Zhang et al., 2022) or an Appendix and also the sample description of the two target groups in two different Tables.

6.       I would differentiate the results according to the segments of patients and caregivers and restructure the Results section in this vein.

7.       I would explain much more in detail what the Colaizzi’s 7-step analysis is (see the best practice example of the paper by Li et al., 2024) and follow the consolidated criteria for reporting qualitative studies (Tong et al. 2007). I would include the reference of the founder of the analysis method (Colaizzi, 1978) in the reference list. The paper by Liu (2019), which was given as the basis for the Colaizzi method cannot be found online. Maybe it was published in Chinese only? The authors should check this reference and if it is only available in Chinese, I would replace it with a similar useful source published in English.

8.       There are some error warnings included in the manuscript (e.g., for linking the Tables to the main text). The authors should check their manuscript in this vein.

9.       After having separated the Results section according to the two perspectives of patients and of caregivers, the Discussion should differentiate between both perspectives and discuss the findings and draw conclusions.

10.   A Limitations section should be added.

11.   Concerning its scientific soundness, the authors should include the two semi-structured interview guidelines for the interviews in a Table or a multimedia appendix.  12.   The methodology description of the study design, the sample recruitment, the data collection, and the data analysis should be significantly expanded. (see also my comments above)

13.   With respect to the readers' interest, the conclusions in general, as well as the practical implications, could be fruitful and interesting. I hope the authors will underline their raison d'être by drawing a clear research gap after having elaborated on what we know about the topic of research of interest, and distinguishing between the two target groups.

14.   In terms of overall merit, I think that in the event that the authors manage to tackle the problems raised, the manuscript under consideration could make a valuable contribution to the research field.

15.   I am not a native speaker, but there are some (minor?) grammatical errors in the manuscript as far as I can judge.

16.   I would not start with the abbreviation of “CVD” in the abstract, but write “Chinese patients suffering from a cardiovascular disease (CVD) and their caregivers…” instead.

The manuscript under consideration can contribute to the research field, but I see some work requiring a diligent revision round. I hope the authors address all raised concerns properly to make this interesting study publishable in Healthcare soon.

Good luck with your research!

References:

Colaizzi, P. (1978). Psychological research as a phenomenologist views it [M]. In Valle, R. & King, M. (Eds.), Existential phenomenological alternatives for psychology (pp. 5–7). Oxford University Press.

Griesser, A., & Bidmon, S. (2024). A holistic view of facilitators and barriers of electronic health records usage from different perspectives: A qualitative content analysis approach. Health Information Management Journal, 53(3), 227-236.

Li, F., Zeng, Y., Fu, Y., Wang, Y., Lin, T., Deng, Q., & Li, J. (2024). Stressors and coping styles of nursing students in the middle period of clinical practicum: a qualitative study. BMC nursing, 23(1), 394.

Liu, M. (2019). Application of Colaizzi’s seven steps in phenomenological research data analysis [J]. Journal of Nursing, 34(11), 90–92. doi:10.3870/j.issn.1001-4152.2019.11.090.

Tong, A., Sainsbury, P., & Craig, J. (2007). Consolidated criteria for reporting qualitative research (COREQ): A 32-item checklist for interviews and focus groups.

Zhang, J. M., Zhang, M. R., Yang, C. H., & Li, Y. (2022). The meaning of life according to patients with advanced lung cancer: a qualitative study. International Journal of Qualitative Studies on Health and Well-being, 17(1), 2028348.

Comments on the Quality of English Language
  1. I am not a native speaker, but there are some (minor?) grammatical errors in the manuscript as far as I can judge.
  2. I would not start with the abbreviation of “CVD” in the abstract, but write “Chinese patients suffering from a cardiovascular disease (CVD) and their caregivers…” instead.

Author Response

(The authors gave the same response as above.)

Round 2

Reviewer 1 Report

Comments and Suggestions for Authors

Dear authors, thank you very much for the very thorough and detailed revision of your manuscript. You have significantly improved it, making it much easier to comprehend.

Author Response

Dear Reviewer,

Thank you very much for your kind and positive feedback on our revised manuscript. We are delighted to hear that you find our revisions thorough and that the manuscript is now significantly improved and easier to comprehend. Your insightful comments and suggestions from the previous round were invaluable in helping us to enhance the clarity and quality of our work.

We have carefully addressed all the points raised and are grateful for the opportunity to have improved our paper accordingly.

Thank you once again for your time and constructive input.

Sincerely,
Zhaoying Zhu
On behalf of all co-authors.

Reviewer 2 Report

Comments and Suggestions for Authors
  • Figure 1 is not readable.
  • The formulas to calculate the ESCA and CPS still not found in manuscript.
  • All other comments are being addressed.

Author Response

Dear Reviewer,

Thank you very much for your thorough review and constructive comments on our manuscript. We sincerely appreciate the time and effort you have dedicated to helping us improve our work. We have carefully considered all your suggestions and have made extensive revisions to the manuscript accordingly. We believe the paper has been significantly strengthened as a result.

Please find below our point-by-point responses to your specific comments in the attached document. All corresponding changes in the manuscript have been marked for your convenience.

Thank you once again for your time and constructive input.

Sincerely,
Zhaoying Zhu
On behalf of all co-authors.

Reviewer 3 Report

Comments and Suggestions for Authors

Dear authors!

Thank you for submitting a revised version of the manuscript entitled “Barriers and facilitators to smart healthcare adoption among Chinese patients with cardiovascular disease and their caregivers: a qualitative study". I appreciate the changes made, and, in my opinion, the quality of the manuscript, which was already high before the former revision round, has improved significantly.

Improvements have been made by elaborating on the rationale supporting the core contribution of the study, focusing on patients with cardiovascular disease (CVD) and their caregivers, and by referencing the existing literature in the field of smart healthcare. You have underlined the research gap, stated a specific research aim, and could underline the raison d'être of your study in this respect. Another significant improvement was achieved through amending the methodology section and restructuring the Results. You have addressed all of my comments satisfactorily.

I appreciate the thorough author’s response and the deliberate addressing of all issues I raised in the previous review round.

To sum up, I believe that the manuscript has improved through revision, and it can make a valuable contribution to the literature in its current form.

Altogether, from my perspective, there remain only two minor issues.

  1. I would invite you to upload a clean version of your manuscript, together with a tracked version of your manuscript, in the future. In its current form, the PDF is difficult to read, as all comments are included next to the main text, which makes readability challenging. Especially, the graphical resolution of Figure 1 is too low so that the boxes included cannot be read at all. I trust, however, that its contents reflect the information included in the main text. Please deliver, additionally, a clean version of a revised manuscript when submitting a revised manuscript in the future. This, however, is not a point of criticism, but a well-intended suggestion from my side.
  2. I would invite a native English speaker to look through the manuscript and especially guarantee the appropriateness of the translation of the verbatim quotes, which have been translated from Mandarin (?) to English. In many sentences, the articles before the nouns are missing. For instance, “I think it’s basically accurate and consistent with hospital doctor” (see lines 793-794) or “I use mobile phone to pay for my family’s medical insurance and hospital fees, which I find very convenient and time-saving.” (see lines 774-775). Some sentences sound odd to me. Some sentences also might have been translated incorrectly with regard to the tenses used, e.g., “I will teach my parents how to use mobile phones to look up questions, and they gradually learned how to do it” (see lines 929-930). From my non-native English perspective, the sentence should start with “I have taught my parents how …” so that it makes sense. A lot of similar sentences are included in the Results section, so I think additional proofreading will be necessary. Typically, translations from the original language to English should follow a specific translation-backtranslation process conducted by bilingual speakers who are fluent in both languages. Please make sure that this was the case and check all of the translations of the embedded verbatim quotes in the manuscript.

Overall, however, I think the manuscript will be publishable after additional proofreading.  I will recommend accepting the present manuscript, provided that it has undergone (another?) round of proofreading.  

The rest of my comments, which I stated in the first review round, have been appropriately addressed, at least from my perspective. I believe the manuscript can make a valuable contribution to the literature in this specific area of research. I hope that I see this interesting manuscript published soon in Healthcare.

Good luck with your research!

Comments on the Quality of English Language

I suggest a thorough proofreading round and additionally invite the authors to check the appropriateness of the translation of the verbatim quotes included in the Results section.  Provided that a translation-backtranslation process with the assistance of a bilingual speaker has been applied to ensure the appropriateness of the correct translation of all verbatim quotes and provided that the entire manuscript has undergone another round of proofreading in general, I would recommend its publication.

Author Response

Dear Reviewer,

Thank you for your review of our revised manuscript and for the highly positive and encouraging feedback. We are delighted to learn that you consider the quality of our manuscript “significant improvement” and believe it “can make a valuable contribution to the literature in this specific area”.

We are particularly grateful for your recognition of the enhancements made in elaborating the theoretical foundation, focusing on the study population, clarifying research gaps and objectives, as well as optimizing the methodology and results sections. Your affirmation means a great deal to us.

We fully agree with your suggestion for additional language proofreading. To ensure the manuscript meets the highest standards of language quality, we have conducted a thorough round of proofreading and verification of the entire text prior to submitting this final version, in accordance with your guidance.

The corresponding certificate of editing and the revised manuscript are attached. We believe that, with this professional language refinement and content improvement, the manuscript now fully complies with the journal’s publication standards.

Once again, we extend our sincere appreciation for your valuable suggestions and enthusiastic encouragement throughout the review process. Your insights have greatly enhanced the quality of our work.

Sincerely,
Zhaoying Zhu
On behalf of all co-authors.
